# The Angiopoietin-2 and TIE Pathway as a Therapeutic Target for Enhancing Antiangiogenic Therapy and Immunotherapy in Patients with Advanced Cancer

**DOI:** 10.3390/ijms21228689

**Published:** 2020-11-18

**Authors:** Alessandra Leong, Minah Kim

**Affiliations:** Department of Pathology and Cell Biology, Columbia University Irving Medical Center, New York, NY 10032, USA; afl2117@cumc.columbia.edu

**Keywords:** ANG2, antiangiogenic therapy, immunotherapy, resistance, cancer treatment

## Abstract

Despite significant advances made in cancer treatment, the development of therapeutic resistance to anticancer drugs represents a major clinical problem that limits treatment efficacy for cancer patients. Herein, we focus on the response and resistance to current antiangiogenic drugs and immunotherapies and describe potential strategies for improved treatment outcomes. Antiangiogenic treatments that mainly target vascular endothelial growth factor (VEGF) signaling have shown efficacy in many types of cancer. However, drug resistance, characterized by disease recurrence, has limited therapeutic success and thus increased our urgency to better understand the mechanism of resistance to inhibitors of VEGF signaling. Moreover, cancer immunotherapies including immune checkpoint inhibitors (ICIs), which stimulate antitumor immunity, have also demonstrated a remarkable clinical benefit in the treatment of many aggressive malignancies. Nevertheless, the emergence of resistance to immunotherapies associated with an immunosuppressive tumor microenvironment has restricted therapeutic response, necessitating the development of better therapeutic strategies to increase treatment efficacy in patients. Angiopoietin-2 (ANG2), which binds to the receptor tyrosine kinase TIE2 in endothelial cells, is a cooperative driver of angiogenesis and vascular destabilization along with VEGF. It has been suggested in multiple preclinical studies that ANG2-mediated vascular changes contribute to the development and persistence of resistance to anti-VEGF therapy. Further, emerging evidence suggests a fundamental link between vascular abnormalities and tumor immune evasion, supporting the rationale for combination strategies of immunotherapy with antiangiogenic drugs. In this review, we discuss the recent mechanistic and clinical advances in targeting angiopoietin signaling, focusing on ANG2 inhibition, to enhance therapeutic efficacy of antiangiogenic and ICI therapies. In short, we propose that a better mechanistic understanding of ANG2-mediated vascular changes will provide insight into the significance of ANG2 in treatment response and resistance to current antiangiogenic and ICI therapies. These advances will ultimately improve therapeutic modalities for cancer treatment.

## 1. Introduction

The vascular network is comprised of blood vessels that act as a crucial delivery system of oxygen and other nutrients to tissues and organs. Healthy blood vessels are lined with endothelial cells that maintain vascular stability and facilitate proper blood flow. Angiogenesis, which is the formation of new blood vessels, plays a vital role in tissue development and maintenance under physiological conditions. However, vascular abnormalities characterized by neovascularization and destabilization are a feature of many diseases including cancer [1,2]. Unlike healthy vasculature, tumor blood vessels are irregularly patterned and morphologically tortuous. Pericytes, which surround healthy blood vessels and maintain vascular stability, are often detached from the endothelial cells of tumor vasculature. Further, impaired endothelial junctions and depleted basement membranes make these tumor vessels unstable. Functionally, this impaired endothelial integrity causes increased vascular leakage that disrupts normal pressure across tumor vessels and promotes dysfunctional flow. The resulting elevated interstitial fluid pressure hinders drug transport into the tumor microenvironment, impeding therapeutic interventions. Additionally, flow stasis in impaired tumor vessels promotes hypoxia, which upregulates angiogenic molecules such as vascular endothelial growth factor-A (VEGF-A, referred to as VEGF), further promoting tumor growth [3]. Together, the structural and functional abnormalities of tumor blood vessels contribute to tumor progression and potentially facilitate the development and persistence of resistance to cancer therapies. Therefore, the reversal of aberrant tumor vasculature, termed “vascular normalization”, serves as a promising target for the treatment of cancer (Figure 1) [3,4].

Tumor vascular remodeling and destabilization are regulated by a set of angiogenic signaling pathways. Indeed, intensive preclinical and clinical studies targeting VEGF/VEGF receptor (VEGFR) signaling, a predominant angiogenic pathway, have shown antiangiogenic and antitumor effects. However, a high incidence of resistance to VEGF signaling-targeted drugs, characterized by disease progression after initial therapy response, has limited the therapeutic success of antiangiogenic drugs in cancer treatment [5,6,7]. This emphasizes the need for improved treatment modalities such as combination treatments with other targeted drugs. Angiopoietin-2 (ANG2) is another predominant angiogenic factor and cooperative driver of vascular destabilization. The ANG2/TIE pathway controls vascular permeability in pathologic conditions and thus acts as a master regulator of vascular stability. Increasing evidence also suggests that vascular destabilization is linked to tumor immune evasion and resistance to current immunotherapy [8,9]. Therefore, targeting the ANG2/TIE pathway in combination with other cancer therapeutics represents a promising clinical strategy for cancer treatment in which vascular stabilization is necessary to increase drug efficacy and inhibit therapeutic resistance. In this review, we focus on recent advances in the understanding of the ANG/TIE pathway in cancer and describe its potential as a therapeutic target to reinforce current antiangiogenic and immune checkpoint inhibitor (ICI) therapies for cancer treatment.

## 2. Angiopoietin/TIE Signaling in Tumor Growth and Metastasis

### 2.1. Current Mechanistic Understanding of Angiopoietin/TIE Signaling

Alongside the VEGF/VEGFR pathway, angiopoietin ligands stimulate angiogenesis and control vascular remodeling and maturation through endothelial TIE receptors, TIE1 (encoded by *TIE1*) and TIE2 (encoded by *TEK*) [10]. The angiopoietin family includes ANG1, ANG2, and ANG4, as well as ANG3, the mouse ortholog of ANG4. Compared with the well-characterized ligands, ANG1 and ANG2, less is known about ANG4 and ANG3, which regulate TIE2 in a species-specific manner [11]. TIE2 is the main binding receptor in the ANG/TIE signaling pathway for all angiopoietins [10,12]. In contrast, the co-receptor TIE1 does not bind any angiopoietins and was long recognized as an orphan receptor until a recent discovery demonstrating leukocyte cell-derived chemotaxin 2 and heparan sulfate glycosaminoglycans as TIE1-ligands [13,14]. Importantly, TIE1 can modulate TIE2 activity and downstream pathway regulation via TIE1/TIE2 receptor dimerization. As such, intact TIE1 and TIE2 expression are required to maintain this pathway and support endothelial barrier integrity [15,16,17].

Pericyte-derived ANG1 is a potent TIE2 activator that supports vascular stability and endothelial barrier function [12,18,19]. Indeed, the respective gene knockout of *ANGPT1* and *TEK* in mice causes embryonic lethality in midgestation as a result of improper vascular remodeling and maturation. Similarly, *TIE1* deletion also contributes to developmental vascular defects, although embryonic lethality results at a later stage [18,19,20]. Mechanistically, in quiescent vasculature, ANG1 binds the TIE2 receptor and constitutively activates it via phosphorylation (p-TIE2). This in turn stimulates the downstream phosphoinositide 3-kinase (PI3K)/AKT singling pathway, leading to AKT phosphorylation (p-AKT) and concomitant inactivation of the forkhead transcription factor FOXO1 (also known as FKHR1). ANG1-mediated inhibition of FOXO1 via nuclear exclusion then promotes the expression of genes involved in vascular stability and suppresses the transcription of factors involved in vascular destabilization, such as ANG2 (Figure 2a) [21,22,23,24,25,26]. 

While ANG1 is a constitutive agonist of TIE2, ANG2 functions as both an agonist and an antagonist of TIE2 in a context-dependent manner. Evidence of ANG2 agonism on TIE2 signaling is most evident in the lymphatic system. Although developmental angiogenesis is largely unaffected in *ANGPT2* knockout mice, ANG2-deficient newborns present with severe lymphatic dysfunction and some mice (of 129/J background) develop postnatal chylous ascites, which leads to postnatal death [27,28,29]. Further, a recent study showed that *ANGPT2* mutations impair TIE1 and TIE2 activation and have been associated with human primary lymphedema, a vascular malformation disorder of the lymphatic system [30]. Other persistent vascular defects such as those found in the retinal and renal vasculature of ANG2-deficient mice suggest the critical role of ANG2 in vessel maturation and maintenance [31,32]. Under vascular quiescence, ANG2 levels are low and ANG1 levels dominate ANG2, which is stored in Weibel–Palade bodies [33]. Under these healthy conditions, ANG2 overexpression induces vascular remodeling without leakage or destabilization, suggesting ANG2 agonism of TIE2 signaling [34]. It has been reported under other conditions that ANG2 can act as a weak agonist of TIE2 [35,36]. Similarly, ANG3 has been reported to function as a TIE2 agonist that promotes corneal angiogenesis. However, overexpression of ANG3 has also been shown to inhibit tumor angiogenesis, indicating context-dependent agonism and antagonism of TIE2 [37,38].

Environmental signals associated with pathogenesis such as hypoxia, VEGF upregulation and increased inflammatory cytokines such as tumor necrosis factor (TNF) can stimulate the release and production of ANG2 in endothelial cells as well as trigger ANG2 antagonism [39,40]. During inflammation, ANG2 competitively suppresses ANG1 activation of TIE2 signaling, causing vessels to become destabilized and leaky [41,42,43,44]. In this context, the extracellular domain of co-receptor TIE1 is cleaved, which is associated with TIE2 inactivation and p-TIE2 suppression [34,45]. PI3K/AKT signaling is subsequently inhibited, leading to the stimulation of FOXO1 activation and ANG2 expression, which engages positive feedback of TIE2 signaling inactivation (Figure 2b) [34]. Infection-induced vascular leakage is also associated with a marked reduction in TIE2 expression levels [46], further compounding pathway inhibition and vascular destabilization. Of note, previous studies have suggested TIE1 acts both positively and negatively on TIE2 in a context-dependent manner [47]. Increasing evidence elucidates the role of TIE1 in angiogenesis and its implications on tumor progression [48,49]. Indeed, a recent study showed that treatment with a TIE1 function-blocking antibody in a presurgical neoadjuvant setting suppressed distant organ metastasis [50], suggesting TIE1 as a promising therapeutic target in cancer treatment. 

Moreover, vascular endothelial protein tyrosine phosphatase (VE-PTP), which is known to associate with VE-cadherin, was shown to suppress TIE2 activation. Interference with VE-PTP stabilizes endothelial junctions via TIE2 by a VE-cadherin-independent mechanism [51]. Another study demonstrated that ANG2, but not ANG1, directly activates β1-integrin to destabilize blood vessels. Here, TIE2 likely acts as a ligand trap to inhibit endothelial ANG2-β1-integrin signaling [52]. A recent study also showed ANG2-β1-integrin signaling in adipose tissue enhances vascular fatty acid transport and prevents peripheral lipid accumulation [53]. Despite the complexity of ANG/TIE signaling regulation, the prominent role of ANG2 and the TIE receptors in the regulation of angiogenesis and vascular destabilization shines a light on the potential of ANG2/TIE signaling as a therapeutic target in cancer treatment. 

### 2.2. The Role of ANG2 in Tumor Growth and Metastasis

As a proangiogenic factor, ANG2 plays a critical role in the growth and metastasis of tumors by modulating vasculature in the tumor microenvironment. Significant upregulation of tissue and blood ANG2 levels has been reported in many cancer types including melanoma, glioblastoma, breast cancer, renal cell carcinoma (RCC) and colorectal cancer and its expression is strongly associated with that of VEGF in angiogenesis and tumor progression [54,55]. A previous study showed the differential expression of ANG2 and VEGF in human tumors, implying tumor specific regulation of expression [56]. Indeed, the prognostic properties of ANG2 in tumor progression have made it a desirable therapeutic target for cancer treatment.

During early stage tumor growth and metastatic development, prior to the angiogenic switch, tumors grow along pre-existing host vasculature in a process known as vessel co-option [57]. During this process, tumorigenic growth activates neighboring endothelial cells which increase ANG2 expression, subsequently driving vessel regression via pericyte depletion of co-opted vessels [58]. ANG2-induced vascular destabilization impairs perfusion and oxygenation of tumor blood vessels, thereby creating a hypoxic niche, which further drives ANG2 increase and the upregulation of other angiogenic factors including VEGF. This process activates the angiogenic switch and initiates tumor angiogenesis to support tumor growth via neovascularization. 

Metastasis is a complex process involving a series of steps and obstacles—primary tumor cell escape, distant organ infiltration, survival and seeding, and ultimate proliferation [59]. Altogether, the process of metastasis is largely inefficient considering the high incidence of tumor cell death prior to successful colonization [60]. ANG2-mediated vascular destabilization in the primary tumor is thought to promote tumor cell intravasation and subsequent migration, facilitating the initial steps of metastasis [61,62,63]. Later, tumor cell extravasation is dependent on both the characteristics of the cancer cells and, importantly, the potential metastatic microenvironment. Specifically, the nature of organ-specific vasculature can facilitate seeding and growth of tumor cells in distant organs. For example, the fenestrated and discontinuous sinusoidal capillaries of the liver and bone marrow are associated with increased metastatic incidence compared to the impermeable blood brain barrier of brain vessels [59]. It is also suggested that the primary tumor can influence future metastatic sites via systemic signaling to promote a premetastatic niche that favors tumor cell extravasation and proliferation in distant organs. The development of supportive niches also relies on activation of proangiogenic cascades through recruitment of TIE2 expressing monocytes (TEMs) and macrophages [64]. Importantly, elevated ANG2 and VEGF levels are associated with a premetastatic niche. Studies in murine melanoma and Lewis lung carcinoma have revealed that increased ANG2 expression in the lung prior to tumor cell arrival as a result of VEGF-mediated activation of the calcineurin-nuclear factor of activated T-cells (NFAT) signaling pathway promotes lung metastasis. This elevated ANG2 expression suggests that ANG2 plays a vital role in the formation of a premetastatic niche [65]. Moreover, a study in melanoma and breast cancer mouse models showed that ANG2 along with matrix metalloproteinase 3 (MMP3) and MMP10 are upregulated in the pulmonary premetastatic niche and are associated with impaired vascular integrity and increased tumor cell extravasation [66]. In both aforementioned studies, ANG2 inhibition, by either a genetic or pharmacological approach, suppressed the formation of lung metastases, confirming the significance of ANG2 in the early development of metastases [65,66]. In addition to enhancing tumor cell extravasation, ANG2 may promote metastatic growth by the subsequent neovascularization of micrometastases via proangiogenic action. Consistently, a study demonstrated that ANG2 inhibition combined with chemotherapy suppresses metastatic growth by limiting the recruitment of tumor-promoting macrophages in the postsurgical adjuvant setting as well as in a preclinical anti-VEGF-refractory tumor model [67]. Another study, consistent with previous reports, found that ANG2 genetic knockout in mice reduced tumor growth in the lungs. However, this deletion of ANG2 unexpectedly increased metastatic growth in the liver as a result of compensatory angiogenic and immunosuppressive mechanisms involving the recruitment of TEMs in tumors [68]. Together, these previous studies emphasize the role of ANG2 in tumor progression and rationalize further investigation into the mechanism of ANG2 action in organ-specific tumor growth and metastasis. 

## 3. Improving Resistance to VEGF Signaling-Targeted Therapy by ANG2 Inhibition

Since the discovery of the VEGF/VEGFR signaling pathway as a master regulator of angiogenesis in development and pathology, clinical translation of anti-VEGF/VEGFR drugs has been extensively implemented in cancer treatment, making antiangiogenic therapies an established standard to combat tumor progression in some advanced cancers including metastatic RCC. The inhibition of new vessel formation by targeting VEGF signaling can retard tumor growth but the emergence of resistance to anti-VEGF agents, which often results in metastatic recurrence, has limited therapeutic success [5,6,7]. Specifically, 20–30% of patients with metastatic RCC do not respond to sunitinib, which targets all VEGF receptors (VEGFRs) and platelet-derived growth factor (PDGF) receptors. Further, patients who initially respond to the treatment often present with disease progression within the next few years. In the adjuvant setting, anti-VEGF drugs also did not demonstrate clinical efficacy as indicated in colorectal cancer [69,70]. Additionally, despite early clinical evidence that bevacizumab, a monoclonal antibody that binds to human VEGF, reduces angiogenesis and tumor burden in human rectal cancer [71], its survival benefit, even when in combination with other anticancer agents, is modest [72]. Preclinical tumor models revealed that VEGF inhibition can paradoxically fuel tumor progression and metastatic spread after initial response to the treatment [5,7]. It is also suggested that the traditional dose of anti-VEGF drugs can cause reduced therapeutic antibody uptake in tumors [73], thus emphasizing the importance of appropriate low-dose antiangiogenic therapy in increasing therapeutic efficacy for cancer treatment [74,75]. Together, understanding the mechanism of response and resistance to inhibitors of VEGF signaling is necessary to improve therapeutic outcomes for cancer patients. 

One proposed mechanism for VEGF-targeted drug resistance suggests that escape mechanisms induce or upregulate other proangiogenic factors such as ANG2 at both the transcriptional and translational levels, consequently desensitizing vessels to VEGF blockade [76]. This suggests the potential for combination treatments designed to inhibit compensatory mechanisms of resistance to improve therapeutic outcomes. Indeed, preclinical and clinical studies have demonstrated the association of ANG2 and VEGF expression in tumors [54,55]. Importantly, a preclinical study showed both ANG2 and TIE2 were upregulated in tumors resistant to VEGFR2-targeted drugs, suggesting that ANG2/TIE2 signaling sustains resistance to VEGF/VEGFR2 inhibitors [77]. Consistently, beneficial antiangiogenic and antitumor effects were observed in multiple preclinical tumor models after combined ANG2 and VEGF blockade [77,78,79,80]. Antitumor efficacy from co-targeting ANG2 and VEGF pathways was also demonstrated in the adjuvant postsurgical setting of preclinical tumor models [81]. Although numerous preclinical studies demonstrated improved antitumor activity when inhibiting both VEGF and ANG2, their combined effects in clinical settings are often inconsistent with preclinical outcomes. For example, a recent phase II clinical trial (NCT02141295), which evaluated the efficacy of vanucizumab (a bispecific ANG2/VEGF-targeting antibody) compared to bevacizumab (anti-VEGF) in metastatic colorectal carcinoma, showed targeting both ANG2 and VEGF did not improve antitumor efficacy [82]. Complexities such as disease stage, tumor type, and treatment dose may account for this observed discrepancy. Of note, another study investigating ANG2 and VEGF double blockade in patients with diabetic macular edema showed a much brighter outcome [83]. Importantly, several ANG2 specific antibodies such as nesvacumab (REGN910) and MEDI3617 have been developed and are currently being tested in combination with other targeted therapies in clinical trials for cancer patients, including those focused on antiangiogenic combination with VEGF inhibitors (Table 1). Previous studies of randomized trials using the ANG1/ANG2 bispecific peptibody, trebananib, in combination with other anticancer agents have resulted in prolonged progression-free survival for patients with recurrent ovarian cancer but not in patients with metastatic colorectal cancer and metastatic RCC [84,85,86]. Considering the opposing roles of ANG1 and ANG2 in the vasculature, the clinical outcomes of targeting ANG2 alone may be more beneficial than from concurrent inhibition of both ANG1 and ANG2.

Further investigation into the mechanism underlying the contribution of ANG2/TIE2 signaling to VEGF-targeted drug resistance will be necessary to identify the potential benefit of combined targeting of ANG2 and VEGF as a treatment modality to overcome therapeutic resistance and to increase the survival benefit for patients. For example, although there is evidence to suggest that ANG2 is involved in the vascular regression of co-opted tumor vessels as well as subsequent regrowth, it remains unknown how ANG2-mediated vascular co-option during tumor progression contributes to anti-VEGF resistance. Additionally, it is unclear whether ANG2 contributes to metastatic tumor growth and thereby promotes anti-VEGF resistance in an organ-specific manner. Previous studies also showed that proinflammatory cytokines such as interleukin (IL)-1β and IL-6 can promote tumor growth and vascular destabilization through modulating the expression of VEGF and ANG2 [87,88]. Upregulation of these proinflammatory cytokines has been also demonstrated in human pancreatic cancer resistant to anti-VEGF treatment. Consistently, these findings show promise in targeting interleukins as potential angiogenesis inhibitors when combined with VEGF and ANG2 inhibitors.

Furthermore, our understanding of ANG2 regulation of TIE signaling has come largely from in vitro experiments and in vivo studies of inflammation [12,35,42,89]. It is therefore necessary to translate the current understanding of ANG2/TIE2 signaling to the tumor microenvironment. Future studies are needed to understand if ANG2 functional switch from TIE2 agonism to antagonism contributes to tumor vascular remodeling and resistance to anti-VEGF therapy during tumor progression. 

## 4. Enhancing Cancer Immunotherapy by Targeting ANG2

### 4.1. Improving Resistance to Immune Checkpoint Inhibitors by ANG2 Inhibition

Cancer immunotherapy, which stimulates antitumor immunity, has shown remarkable efficacy in the treatment of many aggressive malignancies [90,91,92]. Among them, immune checkpoint inhibitors (ICIs), which target inhibitory receptors on T-cells and restore antitumor immune responses, have transformed clinical care for cancer patients. Specifically, treatment with the humanized anti-cytotoxic T lymphocyte antigen 4 (CTLA-4) antibody, ipilimumab, has significantly increased patient survival in advanced cancers [93,94]. The blockade of another immune checkpoint, programmed cell death 1 (PD-1) and its ligand, PD-1 ligand 1 (PD-L1), has demonstrated a survival benefit in a number of different malignancies and has shown higher response rates and reduced side effects compared to anti-CTLA-4 [90,95,96]. Despite the substantial advances in ICI treatment, therapeutic resistance nevertheless limits efficacy in patients. In the case of melanoma, which is particularly susceptible to immunotherapies, approximately half of the patients treated with ICIs show resistance evidenced by disease progression after treatment [97,98].

Many recent studies suggest that the immunosuppressive tumor microenvironment may limit the effectiveness of current immunotherapies including ICIs. Under the current paradigm, an effective response to immunotherapy in solid tumors occurs when tumors are highly inflamed with CD8^+^ T-cells. As such, cold tumors, which lack intratumoral T-cells, tend to present with resistance to immunotherapies [99]. Furthermore, immune exclusion, whereby CD8^+^ T-cells are unable to infiltrate the tumor core from the periphery, is also known to cause resistance to immunotherapy [100]. Various therapeutic strategies have been dedicated to promoting immune cell infiltration into tumors, thereby facilitating an immunosupportive microenvironment and an antitumor immune response [101]. 

The proangiogenic molecule, VEGF, plays an important role in suppressing the native tumor immune response. VEGF/VEGFR2 signaling inhibits dendritic cell maturation and antigen presentation [102] and decreases T-cell maturation and proliferation [103]. Emerging evidence also suggests that angiogenic factors indirectly suppress immunostimulation and facilitate resistance to immunotherapy via vascular destabilization, which consequently enables tumor immune evasion [104,105]. Specifically, increased vascular leakiness and poor vascular perfusion interfere with immune cell trafficking and are suggested to impair the extravasation and infiltration of immune effector cells such as cytotoxic T-cells into tumors [4,8]. Consistently, a study showed that targeting VEGF/VEGFR2 increases the antitumor efficacy of cell transfer therapy by enhancing transferred T-cell infiltration into B16 melanoma [106]. Furthermore, inhibition of VEGF signaling promoted the formation of high endothelial venule (HEV)-like tumor vessels, which facilitated immune cell infiltration and improved response to anti-PD-L1 therapy in a transgenic mouse model for pancreatic neuroendocrine tumors [107]. These recent preclinical studies support the beneficial antitumor effects of targeting VEGF/VEGFR2 signaling in combination with ICIs in mouse tumor models [107,108]. In the clinic, combination of bevacizumab, a humanized monoclonal antibody targeting VEGF, with ipilimumab (anti-CTLA-4) also demonstrated favorable therapeutic outcomes in patients with melanoma compared to ipilimumab treatment alone [109]. Of note, single-agent nivolumab (anti-PD-1) demonstrated an overall survival benefit in patients with metastatic RCC who had progressed after initial antiangiogenic therapy [110]. This result led to FDA treatment approval of nivolumab for metastatic RCC after prior treatment with antiangiogenic therapy. In addition, a recent clinical trial (NCT03434379) in patients with unresectable hepatocellular carcinoma (HCC) demonstrated atezolizumab (anti-PD-L1) combined with bevacizumab resulted in better overall and progression-free survival outcomes than sorafenib [111], leading to recent FDA approval for the treatment of people with unresectable or metastatic HCC who have not received prior systemic therapy. These promising clinical outcomes are consistent with other preclinical studies that demonstrated the beneficial antitumor effects of antiangiogenic therapy in combination with ICIs [107,108]. However, despite intense interest in understanding the effects of vascular normalization on immune cell infiltration to improve ICI therapy efficacy in cancer, it remains unclear if vascular normalization via antiangiogenic therapy can improve treatment efficacy in cases where high CD8^+^ T-cell infiltration does not correlate with response to ICI treatment [112]. Furthermore, hypoxia induction by traditional doses of anti-VEGF drugs presents another challenge for combination treatments with ICIs as hypoxia is known to promote immune suppression in solid tumors. An investigation into the proper doses of antiangiogenic drugs is necessary to promote better treatment outcomes as reduced hypoxia increases the immunosupportive nature of the tumor microenvironment [113]. In fact, a study showed that targeting tumor vasculature with lower doses of anti-VEGFR2 antibody resulted in an immunosupportive tumor microenvironment that potentiated immunotherapy [75]. 

While the role of VEGF signaling in systemic and local immunosuppression has been extensively studied in tumors [4,9,114], little is known about the mechanism underlying the contribution of ANG2/TIE signaling to immune modulation in tumors and to treatment response when combined with ICI therapy. ANG2 exerts immunosuppressive effects by acting directly on immune cells and by promoting the recruitment of TEMs and neutrophils into the tumor microenvironment [115,116]. ANG2 is also known to stimulate TEMs to suppress T-cell activation and to promote the expansion of regulatory T-cell populations [117]. Indirectly, ANG2 can impede immune cell infiltration into tumors by destabilizing tumor blood vessels and consequently disrupting blood flow. Various cytokines including transforming growth factor β (TGFβ), which contribute to immunosuppression, are also implicated with ANG2 and VEGF [8]. For example, IL-6, which exerts immunosuppressive effects through multiple mechanisms including the generation of myeloid-derived suppressor cells [118,119], promotes angiogenesis with defective pericyte coverage through ANG2.

Importantly, ANG2 blockade induces sustained vascular normalization with abundant pericyte coverage around blood vessels and enhanced endothelial junctions whereas VEGF inhibition only induces transient vascular normalization [120]. The dual blockade of VEGF and ANG2 in mice with glioblastoma induced the reprogramming of the immune microenvironment toward immunostimulation [54,80]. A recent study also demonstrated that dual inhibition of VEGF and ANG2 increased antitumor immunity in multiple mouse tumor models including a breast cancer model [108]. In that study, the dual blockade of VEGF and ANG2 increased functional T-cell accumulation, antigen presentation and PD-L1 expression by maximizing vascular normalization. Moreover, recent studies targeting CD40 together with VEGF and ANG2 blockade demonstrated significant benefit in mouse tumor models. One study showed beneficial immune activation after the combined blockade of VEGF and ANG2 with CD40 agonistic antibodies in mice with desmoplastic and chemoresistant colorectal cancer [121]. Similarly, another study showed robust tumor regression associated with increased CD8^+^ T-cell infiltration when anti-CD40 was combined with dual ANG2 and VEGF blockade [122]. Such preclinical studies evaluating the efficacy of dual inhibition of VEGF and ANG2 provide evidence for the simultaneous inhibition of VEGF and ANG2 as a potential treatment strategy to increase the therapeutic efficacy of immunotherapy, including ICIs in cancer treatment (Figure 3) [108,122]. Current clinical trials using ANG2 specific antibodies or an ANG1/ANG2 bispecific peptibody in combination with ICIs will also provide insight into the role of ANG2 in modulating the response to ICIs in cancer patients (Table 1). Further investigation into the mechanism underlying enhanced antitumor immune response by targeting ANG2/TIE2 signaling in combination with ICI therapy without VEGF signaling inhibition will provide a rationale for ANG2 inhibition in combination with ICI therapy as a treatment strategy.

### 4.2. ANG2 as a Potential Biomarker for Therapy With Immune Checkpoint Inhibitors

Despite substantial advances in ICI therapy for cancer treatment, the majority of patients receiving ICIs are not responsive. Therefore, it is necessary to identify and develop predictive biomarkers for patient response to ICI therapy [123]. Clinical designs that allow for individualized therapeutic strategies with ICIs or combination approaches should ideally be led by mechanistic insights to increase therapeutic efficacy and reduce side effects for patients. Currently, three biomarkers are FDA-approved to guide anti-PD-1/PD-L1 therapies: (1) PD-L1 expression [124,125]; (2) mismatch repair deficiency [126] and (3) tumor mutational burden [127,128]. Tumor mutational burden and T-cell-inflamed gene expression profiles are independently predictive of response to ICI therapy [129]. The accumulation of tumor-infiltrating lymphocytes (TILs) in tumors has also been considered as another potential predictor of response to ICI therapy. For PD-1 therapy, the number of TILs at the margin of tumors is strongly associated with treatment response [130]. Consistently, a recent clinical trial evaluating the efficacy of atezolizumab plus bevacizumab or sunitinib as a first-line treatment for clear-cell RCC showed that the combination treatment resulted in a notable improvement in patients with high numbers of TILs and high levels of PD-L1 expression in immune cells [131]. In addition, treatment with pembrolizumab (anti-PD-1) is also approved for patients with non-small-cell lung cancer (NSCLC) who have high PD-L1 expression on tumor cells (at least 50% of tumor cells for first-line use). Nevertheless, PD-L1 expression measured through immunohistochemistry has proved insufficient to predict ICI response in patients due to contradictory results in recent studies [110,132]. 

Unlike conventional tissue biopsies, clinical biomarkers are ideally assessed via less invasive manners. Accordingly, intensive ongoing work has been dedicated to developing blood-derived systemic biomarkers for immunotherapy response. For example, several studies suggest the association of peripheral blood composition, including total lymphocyte count, T-cell clonality and cytokine levels, with ICI response [133,134]. Additionally, an increase in circulating exosomal PD-L1 during treatment was shown to correlate with response to anti-PD-1 therapy in patients with melanoma [135]. Given increasing evidence, which suggests that angiogenic factors play a part in regulating immune function [104,136], circulating angiogenic molecules may serve as potential biomarkers for predicting immune status and response to treatment with immunotherapy. Indeed, pretreatment serum with higher VEGF levels has shown to be associated with poor survival in metastatic melanoma patients treated with ipilimumab, providing a strong rationale for targeting VEGF signaling with ICIs [137]. Interestingly, high pretreatment levels of serum VEGF-C were associated with improved progression-free survival in melanoma patients treated with combined inhibition of CTLA-4 and PD-1 [138]. 

High circulating levels of ANG2 have been linked to poor prognosis, disease progression and metastasis in many types of advanced tumors [10,139,140,141], highlighting the importance of ANG2 as a biomarker and tool to understand cancer progression. Circulating ANG2 levels have also been associated with disease progression in metastatic melanoma [142] and were identified as a prognostic biomarker in colorectal cancer treated with a VEGF inhibitor [143]. Of note, a recent study showed that increased levels of serum ANG2 are correlated with poor response in patients with metastatic melanoma treated with ipilimumab [144]. Interestingly, when ipilimumab was combined with bevacizumab, ANG2 levels decreased in a few patients but not in others, suggesting the potential of targeting VEGF together with ANG2 in these patients. Further investigation focused on ANG2 efficacy alone or in combination with other biomarkers is necessary to validate ANG2 as a predictive biomarker for ICI response in different cancer types. Successful outcomes will allow for individualized treatment that maximizes therapeutic efficacy for cancer patients.

## 5. Conclusions

Recent preclinical and clinical studies provide a rationale for ANG2 as a potential candidate for therapeutic interventions in cancer treatment for a range of solid tumors. Indeed, several drugs targeting ANG2/TIE2 signaling are currently under clinical trials as combination therapies. However, the complexities of ANG2 function as a TIE2 agonist/antagonist along with the complex interplay of ANG2 with ANG1 require careful investigation for the clinical development of drugs that target ANG2, both ANG2 and ANG1, or TIE receptors. The vascular stabilizing effects of ANG2 inhibition on delaying tumor progression and stimulating antitumor immunity make ANG2 an ideal target for combination treatment with other targeted therapies, especially those that induce resistance. Herein, we discussed the potential for combined ANG2 inhibition with current antiangiogenic therapy or ICIs to increase therapeutic outcomes in cancer patients. Antiangiogenic treatments that primarily target VEGF signaling have shown efficacy in cancer treatment but have generally failed to improve the overall survival of patients in most cancers. ICI therapy, a potent immunotherapy, has shown remarkable efficacy in many advanced malignancies by demonstrating durable responses and prolonged survival; however, ICI treatment also failed to improve therapeutic outcomes for the majority of patients. ANG2 inhibitors are currently being tested in different clinical trials in combination with antiangiogenic drugs or with ICI therapy in cancer patients. Further mechanistic studies are necessary to provide a better understanding of how ANG2 differentially contributes to the development and persistence of resistance to anti-VEGF and ICI therapies with respect to cancer type and tumor stage. This mechanistic understanding will provide insight into better therapeutic strategies to improve treatment efficacy in cancer patients. Moreover, future work is needed to determine if ANG2 alone or in combination with other biomarkers can serve as better predictive biomarkers for antiangiogenic and ICI therapies.

## Figures and Tables

**Figure 1 ijms-21-08689-f001:**
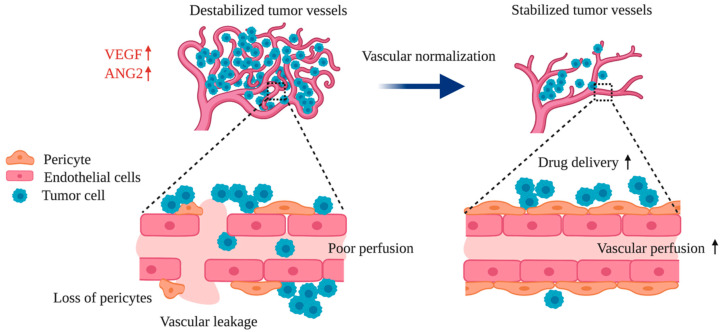
Vascular destabilization and normalization in the tumor microenvironment. Vascular endothelial growth factor-A (VEGF) and angiopoietin-2 (ANG2) levels are upregulated in the tumor microenvironment where vessels are functionally and structurally altered. Destabilized tumor vessels are characterized by pericyte loss, impaired endothelial junctions, increased vascular leakage and poor vascular perfusion. Vascular normalization by VEGF and/or ANG2 blockade stabilizes tumor vessels and induces increased vascular perfusion, which consequently improves drug delivery in cancer treatment.

**Figure 2 ijms-21-08689-f002:**
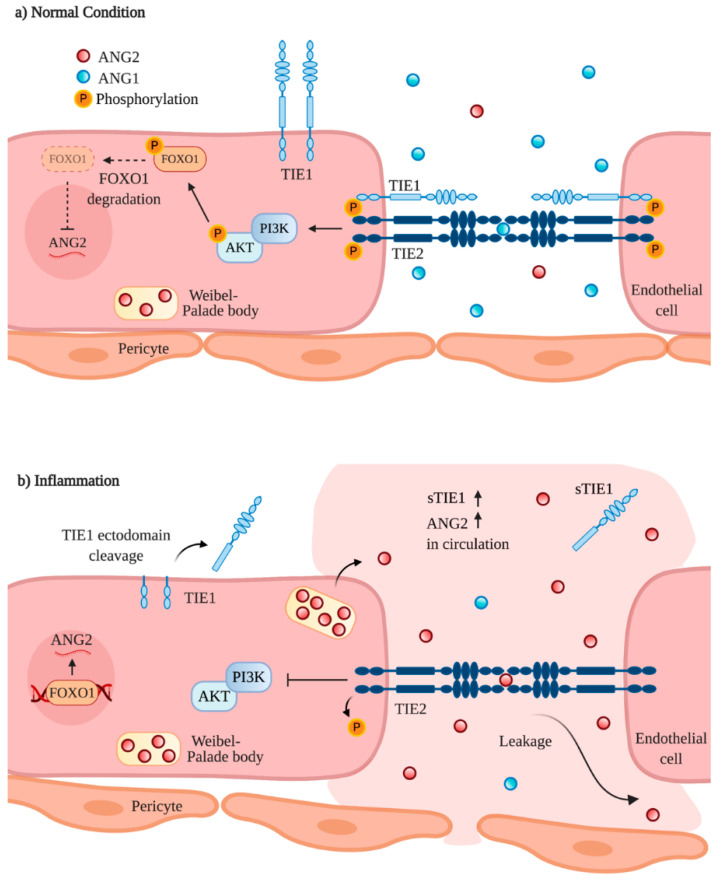
Angiopoietin-TIE signaling under normal conditions and inflammation. (**a**) Under normal conditions, when ANG2 expression is low, ANG2 is stored in Weibel–Palade bodies. ANG1, which dominates ANG2 levels, binds to and activates TIE2. Along with co-receptor TIE1, TIE2 is phosphorylated (p-TIE2), which in turn activates phosphoinositide 3-kinase (PI3K/AKT) signaling. Phosphorylated AKT inactivates the forkhead transcription factor (FOXO1) via nuclear exclusion, inhibiting further ANG2 expression. As a result, vessel quiescence and vascular stability are maintained. (**b**) In inflammation, ANG2 is upregulated and competes with ANG1 for TIE2 receptor binding. The ectodomain of co-receptor TIE1 is cleaved leading to p-TIE2 suppression and PI3K/AKT inactivation. Consequently, activated FOXO1 increases ANG2 expression, which sustains positive feedback of TIE2 signaling suppression and ultimately contributes to vascular destabilization and leakage.

**Figure 3 ijms-21-08689-f003:**
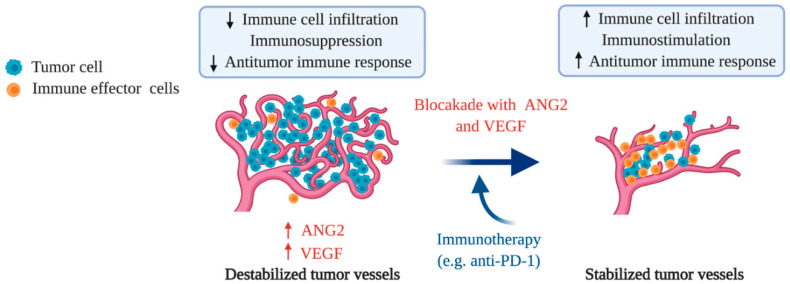
Immunostimulation and enhanced antitumor efficacy of immunotherapy by vascular stabilization. The destabilized vasculature in tumors impairs infiltration of immune effector cells (such as CD8^+^ T-cells) into the tumor microenvironment, which consequently induces immunosuppression. Vascular stabilization, by targeting the antiangiogenic molecules, ANG2 and VEGF, promotes immune cell infiltration and immunostimulation. Here, immunotherapies such as immune checkpoint inhibitors (ICIs) can further activate suppressed immune effector cells to induce a robust antitumor immune response. This combination treatment modality of antiangiogenic drugs with immunotherapy maximizes therapeutic efficacy in cancer treatment, especially during advanced stage disease.

**Table 1 ijms-21-08689-t001:** Clinical studies testing ANG-targeted therapies in combination with antiangiogenic and immunotherapies in cancer treatment.

Clinicaltrials.gov ID	ANG Blockade	Combination Treatment	Tumor Type	Phase and Status	Competitor Arm
***Combined with Immunotherapies***					
NCT03239145	Trebananib (ANG1 and ANG2)	Pembrolizumab (PD-1)	Melanoma, RCC, ovarian cancer, colorectal cancer	Phase 1: recruiting	
NCT01688206	Vanucizumab (VEGF-A and ANG2)	Atezolizumab (PD-L1)	Solid tumors	Phase 1: completed	
NCT02665416	Vanucizumab	Selicrelumab (CD40)	Solid tumors	Phase 1: completed	Bevacizumab + Selicrelumab
NCT02141542	MEDI3617 (ANG2)	Tremelimumab (CTLA-4)	Metastatic melanoma	Phase 1: active, not recruiting	
***Combined with anti-VEGF/VEGFR2 drugs***					
NCT01688960	Nesvacumab (ANG2)	Aflibercept (VEGF, VEGF-B, and PIGF)	Solid tumors	Phase 1: completed	
NCT01664182	Trebananib	Bevacizumab (VEGF-A)	Advanced renal cell carcinoma	Phase 2: active, not recruiting	Trebananib + Pazopanib Hydrochloride; Trebananib + Sorafenib Tosylate; Trebananib + Sunitinib Malate
NCT01609790	Trebananib	Bevacizumab	Glioma	Phase 2: active, not recruiting	
NCT01290263	Trebananib	Bevacizumab	Glioblastoma	Phase 1 and Phase 2: completed	
NCT01249521	Trebananib	Bevacizumab	Metastatic colorectal cancer	Phase 2	
NCT02597036	LY3127804 (ANG2)	Ramucirumab (VEGFR2)	Solid tumors	Phase 1: active, not recruiting	LY3127804 + Ramucirumab + Paclitaxel

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
