# Peer review of "The Angiopoietin-2 and TIE Pathway as a Therapeutic Target for Enhancing Antiangiogenic Therapy and Immunotherapy in Patients with Advanced Cancer"

_ijms, 2020, doi:10.3390/ijms21228689_

Round 1
Reviewer 1 Report
The authors provide a solid overview of the angiopoietin-Tie pathway, and discuss its role in cancer progression as well as its potential as a therapeutic target to overcome resistance of current anti-angiogenic and immuno-therapy regimens. Overall, this review is timely, logically structured, and covers the clinical potential of Ang2-targeting in a comprehensive manner. Some changes may be warranted to to improve the overall quality of this review. The authors are particularly encouraged to consider the following:
1. Page3, line No.82-83: The authors protrait Tie1 as an “orphan receptor to which no known ligands directly bind”. While this statement was true for many years, LECT2 (Xu et al., Cell 2019) and heparan sulfate glycosaminoglycans (Griffin et al., Nature 2020) have recently been identified as Tie1-ligands. Please revise!
2. Page3: The authors nicely outline the importance of Ang2 for lymphatic development in mice. The recent paper by the group of Kari Alitalo showing that Ang2 mutations are associated with primary lymphedema in humans is likely relevant in this context and should be discussed (Leppänen et al., Sci. Transl. Med. 2020).
3. Page 5, line No.180: “A recently study” should be corrected to “A recent study”.
4. Page 6, line No.210-211: Do the authors mean post-translational rather than “post-transcriptional” protein modifications?
5. Page 6, line No.215-228: The authors comprehensively discuss the preclinical evidence suggesting that combined Ang2 and VEGF neutralization would be superior to anti-VEGF monotherapy. However, this is only true in the preclinical setting in which the monodrugs were given at suboptimal concentrations to see the synergistic effect. Clinically, this – sadly – did not translate and truth of the matter is that essentially all pharmaceutical companies have stopped their Angpt2 clinical program in oncology (Roche still pursuing this in combination with IO). Thus, the authors are strongly encouraged to discuss the translational ramifications somewhat more realistically. The report of the failed CRC clinical trial (Bendell et al, Oncologist 2020) that compared the efficacy of a bispecific Ang2/VEGF-targeting antibody (Vanucizumab) to Bevaciczumab treatment should be included. While this review focusses on oncology, the authors may want to make reference to the much brighter outlook of Ang2/VEGF double targeting in the field of ophthalmology (phase 2 published – phase 3 ongoing). But here, the issue is not necessarily efficacy but durability (16 weeks vs. 8 weeks).
6. Page 8, line No. 282-300: The recently completed ImBrave-Trial showing that the combination of anti-VEGF and anti-PD-L1 (Atezolizumab) is superior to Sorafenib in advanced hepatocellular carcinoma should be included in the discussion (Finn et al, N. Engl. J. Med. 2020). This study is in so far remarkable that Atezolizumab had previously failed as a monotherapy in HCC. And future studies will likely need to show if Angpt2 can add something to this.
7. Page 9, line No.319: Two recent studies showed augmented CD40 immunostimulatory effects in combination with VEGF/Ang2 targeting (Simone Ragusa et al., J. Clin. Invest. 2020; Abhishek et al., Kashyap, PNAS 2020). Both papers should be discussed and cited.
8. While the manuscript comprehensively discusses Ang2 as a potential target to enhance immune checkpoint blockade and anti-angiogenic therapy, it neglects a number of recent preclinical studies that suggested Tie1 as an additional promising target within the Ang-Tie pathway for tumor therapy (Singhal et al, EMBO Mol Med 2020; La Porta et al, J Clin Invest 2018; D’Amico et al, J Clin Invest 2014).
Author Response
Comments and Suggestions for Authors
Reviewer 1
The authors provide a solid overview of the angiopoietin-Tie pathway, and discuss its role in cancer progression as well as its potential as a therapeutic target to overcome resistance of current anti-angiogenic and immuno-therapy regimens. Overall, this review is timely, logically structured, and covers the clinical potential of Ang2-targeting in a comprehensive manner. Some changes may be warranted to to improve the overall quality of this review. The authors are particularly encouraged to consider the following:
We thank reviewers for the favorable assessment and for their insightful and thoughtful comments. We have addressed these comments in the revised manuscript by incorporating suggestions and citations to improve the timeliness and relevancy of the review.
1. Page3, line No.82-83: The authors protrait Tie1 as an “orphan receptor to which no known ligands directly bind”. While this statement was true for many years, LECT2 (Xu et al., Cell 2019) and heparan sulfate glycosaminoglycans (Griffin et al., Nature 2020) have recently been identified as Tie1-ligands. Please revise!
: We thank the reviewer for drawing our attention to the important published findings. In our revised manuscript, we corrected the description of TIE1 and now highlight the recently discovered ligands (Page 3, lines 85-88) with proper citations.
2. Page3: The authors nicely outline the importance of Ang2 for lymphatic development in mice. The recent paper by the group of Kari Alitalo showing that Ang2 mutations are associated with primary lymphedema in humans is likely relevant in this context and should be discussed (Leppänen et al., Sci. Transl. Med. 2020).
: We agree with the reviewer’s comment. The relevant paper by Leppänen et al. has been included in the revised manuscript (Page 3, lines 107-109) to strengthen our discussion of ANG2 agonism in development and lymphatics, particularly in human physiology.
3. Page 5, line No.180: “A recently study” should be corrected to “A recent study”.
: We thank the reviewer for catching this error. It has been corrected in the revised manuscript.
4. Page 6, line No.210-211: Do the authors mean post-translational rather than “post-transcriptional” protein modifications?
: We thank the reviewer for pointing out an inadequate description. To clarify the sentiment, we rephrased the sentence (Page 6, lines 228-230) and added a proper citation. Resistance mechanisms of VEGF inhibitor are proposed at both transcriptional and translational levels to increase angiogenic factors and ultimately mount resistance responses. This idea has been more thoroughly addressed in the revised manuscript.
5. Page 6, line No.215-228: The authors comprehensively discuss the preclinical evidence suggesting that combined Ang2 and VEGF neutralization would be superior to anti-VEGF monotherapy. However, this is only true in the preclinical setting in which the monodrugs were given at suboptimal concentrations to see the synergistic effect. Clinically, this – sadly – did not translate and truth of the matter is that essentially all pharmaceutical companies have stopped their Angpt2 clinical program in oncology (Roche still pursuing this in combination with IO). Thus, the authors are strongly encouraged to discuss the translational ramifications somewhat more realistically. The report of the failed CRC clinical trial (Bendell et al, Oncologist 2020) that compared the efficacy of a bispecific Ang2/VEGF-targeting antibody (Vanucizumab) to Bevaciczumab treatment should be included. While this review focusses on oncology, the authors may want to make reference to the much brighter outlook of Ang2/VEGF double targeting in the field of ophthalmology (phase 2 published – phase 3 ongoing). But here, the issue is not necessarily efficacy but durability (16 weeks vs. 8 weeks).
: We thank the reviewer for the insightful comment and agree with the reviewer that a more complete picture of available clinical results should be included. To address this concern, in the revised manuscript, we now include a direct discussion of the translation of combination treatments with ANG2 and VEGF in clinical trials. We have included a full discussion of the failed clinical trial with Vanucizumab and Bevaciczumab along with results (Page 6, lines 239-245) and proper citation. Additionally, to provide a well-rounded picture of therapeutic ANG2 and VEGF combination efficacy, we included positive results from a trial targeting dual blockade of ANG2 and VEGF in diabetic macular edema (Page 6, lines 245-247).
6. Page 8, line No. 282-300: The recently completed ImBrave-Trial showing that the combination of anti-VEGF and anti-PD-L1 (Atezolizumab) is superior to Sorafenib in advanced hepatocellular carcinoma should be included in the discussion (Finn et al, N. Engl. J. Med. 2020). This study is in so far remarkable that Atezolizumab had previously failed as a monotherapy in HCC. And future studies will likely need to show if Angpt2 can add something to this.
: We thank the reviewer for bringing the ImBrave-Trial to our attention. We agree with the reviewer on the relevancy of the study to our manuscript and have added a discussion of the trial in the revised manuscript (Page 9, lines 320-324) to strengthen our review.
7. Page 9, line No.319: Two recent studies showed augmented CD40 immunostimulatory effects in combination with VEGF/Ang2 targeting (Simone Ragusa et al., J. Clin. Invest. 2020; Abhishek et al., Kashyap, PNAS 2020). Both papers should be discussed and cited.
: We thank the reviewer for including two recent and relevant papers on combination immunotherapies with antiangiogenic drugs. A discussion of the two indicated papers have been discussed in the revised manuscript and cited as well (Page 9, lines 359-364).
8. While the manuscript comprehensively discusses Ang2 as a potential target to enhance immune checkpoint blockade and anti-angiogenic therapy, it neglects a number of recent preclinical studies that suggested Tie1 as an additional promising target within the Ang-Tie pathway for tumor therapy (Singhal et al, EMBO Mol Med 2020; La Porta et al, J Clin Invest 2018; D’Amico et al, J Clin Invest 2014).
: We thank the reviewer and agree with the reviewer’s comment. As suggested by the reviewer, we have included a discussion of the therapeutic potential of TIE1 as a target in cancer treatment (Page 3-4, lines 127-132) and cited the suggested papers to provide an enhanced picture of the therapeutic potential of targeting ANG/TIE signaling. Further, in our revised manuscript, a deeper description of TIE1 action in ANG/TIE signaling has been addressed with relevant citations (Page 3, lines 85-90 and 94-95).
Reviewer 2 Report
The reviewed manuscript is interesting. The authors have described the latest findings in angiopoietin-2 and TIE2 pathway field. Especially, they have discussed the role of ANG/TIE2 pathway in cancer and described its potential as a therapeutic target to reinforce current anti-VEGF drugs and immune checkpoint inhibitor therapies for cancer treatment.
However, the manuscript should be improved in several points before publication:
Please introduce brief information about the angiopoietin family and add a table with comparative characteristic of all members of the angiopoietin family.
The authors should also provide comparative information about changes in expression level of VEGF, Ang2 and Tie2 in different types of cancer.
Please add information about relation between VEGF/Ang2/Tie2 pathway and various cytokine pathways. Please note that it is important to discuss research results about impact of cytokines on VEGF/Ang2/Tie2 pathway especially if the authors discussed and presented figure about immunosuppression and immunostimulant effects (e.g., figure 3).
What kind of interleukins could be used as potential angiogenesis inhibitors through VEGF/Ang2/Tie2 pathway in cancers .
Author Response
Reviewer 2
The reviewed manuscript is interesting. The authors have described the latest findings in angiopoietin-2 and TIE2 pathway field. Especially, they have discussed the role of ANG/TIE2 pathway in cancer and described its potential as a therapeutic target to reinforce current anti-VEGF drugs and immune checkpoint inhibitor therapies for cancer treatment. However, the manuscript should be improved in several points before publication:
We thank reviewers for the favorable assessment and for their insightful and thoughtful comments. We have addressed these comments in the revised manuscript by incorporating suggestions and citations to improve the timeliness and relevancy of the review.
1. Please introduce brief information about the angiopoietin family and add a table with comparative characteristic of all members of the angiopoietin family.
: We thank the reviewer for the comment. To provide more context to ANG/TIE signaling, we have included a discussion of the full angiopoietin family and provided more information about family members, ANG3 and ANG4. The comparative functions of ANG3 and ANG4 are discussed in the revised manuscript (Page 3, lines 82-85, 115-117) with descriptions of context dependent functions and interactions with TIE receptors. Descriptions of ANG3 and ANG4 are incorporated into the revised manuscript alongside detailed discussions of ANG2 and ANG1 for context and comparison.
2. The authors should also provide comparative information about changes in expression level of VEGF, Ang2 and Tie2 in different types of cancer.
: We thank the reviewer for the comment and agree with the reviewer’s suggestion to provide deeper context in the relationship between ANG2 and VEGF in cancer. In the revised manuscript (Page 5, lines 156-161), we draw attention to the association of ANG2 and VEGF in cancer as well as their increased expression in tumor progression. Importantly, we include a citation (#61, Hashizune et al.) quantifying the comparative gene expression of ANG2, VEGF, and TIE2 in different human cancers. This information provides insights into the tumor specific regulation of ANG/TIE signaling as well as insight into which certain cancers may be strong candidates for targeted interventions.
3. Please add information about relation between VEGF/Ang2/Tie2 pathway and various cytokine pathways. Please note that it is important to discuss research results about impact of cytokines on VEGF/Ang2/Tie2 pathway especially if the authors discussed and presented figure about immunosuppression and immunostimulant effects (e.g., figure 3).
: We thank the reviewer for the comments and agree a discussion of cytokine involvement with ANG2 and VEGF signaling will strengthen our discussion of angiogenic process associated with immunosuppression. To address this comment, we have included references for the involvement and impact of cytokines such as interleukins on angiogenic processes involving ANG2 and VEGF and tumor immunosuppression (Page 9, lines 346-350).
4. What kind of interleukins could be used as potential angiogenesis inhibitors through VEGF/Ang2/Tie2 pathway in cancers
: We thank the reviewer for the insightful inquiry. We have included comments on interleukins, IL-1β and IL-6, with tumorigenic and angiogenic potential that act via ANG2 and VEGF (Page 7, lines 263-268). We suggest that targeting interleukins, which promote angiogenesis and vascular destabilization, present promise as potential angiogenesis inhibitors alongside ANG2 and VEGF inhibition.
Round 2
Reviewer 2 Report
The authors have improved manuscript according to my comments